# Predictive Roles of ADAM17 in Patient Survival and Immune Cell Infiltration in Hepatocellular Carcinoma

**DOI:** 10.3390/ijms242317069

**Published:** 2023-12-02

**Authors:** Tianlong Ding, Yang Yu, Lei Gao, Lin Xiang, Bo Xu, Baohong Gu, Hao Chen

**Affiliations:** 1The Department of Tumor Surgery, Lanzhou University Second Hospital, Lanzhou 730030, China; gxlz8002@163.com; 2The Second Clinical Medical College, Lanzhou University, Lanzhou 730030, China; yuy2021@lzu.edu.cn (Y.Y.); 18690484801@163.com (L.G.); xiangl18@lzu.edu.cn (L.X.); ldxb0218@126.com (B.X.); gubh19@lzu.edu.cn (B.G.)

**Keywords:** ADAM17, hepatocellular carcinoma, immune cell infiltration, prognosis, proliferation, metastasis, immune evasion, metalloproteinase, signalling pathway

## Abstract

Hepatocellular carcinoma (HCC) is the deadliest malignant tumour worldwide. The metalloproteinase ADAM17 is associated with tumour formation and development; however, its significance in HCC is unclear. This study aimed to investigate the role of ADAM17 in HCC and the correlation between its expression and immune cell infiltration. ADAM17 expression was analysed in pan-cancer and HCC tissues using The Cancer Genome Atlas and Genotype-Tissue Expression datasets. Kaplan–Meier survival analysis displayed a negative association between ADAM17 expression and the overall survival of patients with HCC. High ADAM17 expression was linked to poor tumour/node (T/N) stage and alpha fetoprotein (AFP) levels. Gene Set Enrichment Analysis, Gene Ontology, and Kyoto Encyclopaedia of Genes and Genomes analyses revealed the enrichment of several pathways, including epithelial–mesenchymal transition, inflammatory response, Hedgehog, and KRAS signalling, in patients with upregulated ADAM17. ADAM17 was shown to be positively correlated with immune cell infiltration and immune checkpoint expression via the Tumour Immune Estimation Resource (TIMER) database and immunohistochemistry analyses. Protein–protein interaction (PPI) network analysis revealed that ADAM17 plays a core role in cancer development and immune evasion. In vitro and in vivo experiments demonstrated that ADAM17 influences HCC growth and metastasis. In conclusion, ADAM17 is upregulated in most cancers, particularly HCC, and is critical in the development and immune evasion of HCC.

## 1. Introduction

Cancer is one of the most serious diseases affecting human health worldwide. Liver cancer is the sixth most common malignancy and the third leading cause of cancer-related death worldwide [1]. Hepatocellular carcinoma (HCC) is the most common pathological type of liver cancer, accounting for approximately 90% of all liver cancer cases [2]. These tumours have a high propensity for recurrence and metastasis, resulting in an unsatisfactory prognosis in patients with HCC who undergo radical resection [3,4]. The American Joint Committee on Cancer (AJCC) TNM staging system is particularly important in the surgical resection and prognosis evaluation of HCC. According to the AJCC cancer staging criteria, T, N, and M represent the primary tumour, regional lymph nodes, and distant metastasis, respectively. Combined with the categories of T, N, and M, the AJCC proposed prognostic stage groups [5]. Recently, targeted therapy and immunotherapy have greatly improved HCC treatment. However, a limited number of patients benefit from these treatments because of drug insensitivity or resistance [6,7]. It is, therefore, paramount to explore the critical events in cancer development and further elucidate the molecular mechanisms behind HCC progression, to aid in the discovery of novel and potential therapeutic targets.

ADAM17, also known as the tumour necrosis factor-alpha converting enzyme (TACE), is a transmembrane cell surface metalloproteinase that is widely expressed in various cell and tissue types [8,9,10,11]. The main body of the ADAM17 enzyme is composed of a cysteine-rich membrane proximal domain, transmembrane domain, cytoplasmic domain, prodomain, catalytic domain, and disintegrin domain [12]. Numerous studies have confirmed that the shedding activity of ADAM17 is enhanced in many tumours, causing ADAM17 to be abnormally expressed [11]. This observation has been noted in breast [13], lung [14,15], gastric [16], and colorectal cancers [17], pancreatic adenocarcinoma [18], and head and neck squamous cell carcinoma [19]. The tumour microenvironment (TME) is a complex environment composed of a variety of cells, extracellular matrix, and signalling molecules [20] and is closely related to tumour cell proliferation, drug resistance, immune evasion, metastasis, and angiogenesis [21,22]. Multiple studies have shown that ADAM17 participates in the immune regulation of tumours and plays an important role in their formation and development [23,24].

In this study, we explored the association between ADAM17 expression and the clinical features of HCC, as well as the role of ADAM17 in HCC prognosis. The enriched signalling pathways, gene profiles, and protein–protein interaction (PPI) networks of ADAM17 in HCC were examined, and the association between ADAM17 expression and immune cell infiltration was evaluated. The biological functions of ADAM17 in HCC were investigated in vitro and in vivo.

## 2. Results

### 2.1. Expression of ADAM17 in Pan-Cancer and HCC

The expression of ADAM17 was first explored in pan-cancer using The Cancer Genome Atlas (TCGA) cohort. As shown in Figure 1A,B, ADAM17 was overexpressed in most tumour tissues compared to normal tissues, including bladder (BLCA), cervical (CESC), bile duct (CHOL), colon (COAD), oesophageal (ESCA), head and neck (HNSC), liver (LIHC), rectal (READ), stomach (STAD), and endometrioid (UCEC) cancers; glioblastoma (GBM); and kidney clear cell (KIRC) and lung squamous cell (LUSC) carcinomas. Specific to HCC, we analysed the level of ADAM17 in TCGA-LIHC cohorts, including or excluding the Genotype-Tissue Expression (GTEx) datasets. Boxplots showed that ADAM17 expression increased significantly in HCC tissues compared to that in normal tissues (Figure 1C,D).

### 2.2. Prognostic Role of ADAM17 and Its Correlation with HCC Clinicopathological Features

Survival analysis of patients with HCC showed that patients with high ADAM17 expression had significantly poorer OS than those with low ADAM17 expression (Figure 1E,F). These findings indicate the prognostic significance of ADAM17 expression in patients with HCC. In addition, we explored the correlation between ADAM17 expression and the clinicopathological features of patients with HCC. Boxplots showed that the increase in ADAM17 was associated with poor T and N stages of cancer, where T refers to the size and extent of the main tumour and N refers to the number of nearby lymph nodes affected. Moreover, ADAM17 expression was positively correlated with the level of alpha-fetoprotein (AFP; Figure 2A–D). Patient characteristics included in these analyses are presented in Table 1.

### 2.3. Investigating the Function of ADAM17 and Related Pathways

Differentially expressed genes (DEGs) were identified in patients with high and low ADAM17 expression levels (Figure 3A). These DEGs were subjected to enrichment analysis. Gene Set Enrichment Analysis (GSEA) based on Kyoto Encyclopaedia of Genes and Genomes (KEGG) pathway analysis revealed that the hallmark pathways of epithelial–mesenchymal transition (EMT), inflammatory response, Hedgehog signalling, and KRAS signalling UP were significantly enriched in the high ADAM17 expression group (Figure 3B). Gene Ontology (GO) annotation and KEGG pathway analyses were performed to further understand the function of ADAM17. The ADAM17-related DEGs were enriched in the following top terms (Figure 3C): cell–cell adhesion via plasma–membrane adhesion, molecules external encapsulating structure organisation, homophilic cell adhesion via plasma membrane adhesion molecules (BP); collagen-containing extracellular matrix, transporter complex, transmembrane transporter complex (CC); passive transmembrane transporter activity, channel activity, extracellular matrix structural constituent (MF); neuroactive ligandreceptor interaction, oxidative phosphorylation, and extracellular matrix (ECM)-receptor interaction (KEGG).

### 2.4. Association between ADAM17 Expression, Immune Infiltration, and Copy Number

The TME plays an important role in cancer development and immune escape. First, we analysed the correlation between ADAM17 levels and immune cell infiltration using the Tumour Immune Estimation Resource (TIMER) database. Using the TIMER algorithm, six types of immune cells were identified in the TME. Correlation analyses revealed that ADAM17 was negatively correlated with the purity of the tumour, but positively correlated with the levels of six immune cells, including B cells, CD8+ T cells, CD4+ cells, macrophages, neutrophils, and dendritic cells (Figure 4A). Furthermore, we analysed the relationship between ADAM17 and the immune checkpoints. The results showed that the expression of ADAM17 was positively correlated with the levels of immune checkpoints in the immune microenvironment (Figure 4B). In addition, the copy number of ADAM17 was found to be related to immune cell infiltration (Figure 4C). These results indicate that ADAM17 can affect immune cell infiltration and create an immunosuppressive microenvironment.

### 2.5. Gene Profile Analysis of ADAM17

We analysed gene mutations in ADAM17 using the online cBioPortal tool. The mutation status of ADAM17 in pan-cancer cells is shown in Figure 5A. In HCC, ADAM17 has a relatively high mutation rate, with approximately 3% of patients harbouring genetic alterations (Figure 5A,C). We also found a relationship between the mutation and ADAM17 mRNA expression (Figure 5B), highlighting the role of gene alteration in ADAM17-driven tumorigenesis.

### 2.6. PPI Network Analysis

The top genes that directly interacted with ADAM17 were identified, and a network was built. As shown in Figure 6A, several oncogenes and immune genes were involved in this network. This further confirms the important role of ADAM17 in cancer development and immune evasion. Enrichment analysis based on these genes identified the following terms: membrane protein proteolysis, membrane protein ectodomain proteolysis, Notch signalling pathway, membrane microdomain, membrane raft, apical part of the cell, signalling receptor activator activity, endopeptidase activity, growth factor receptor binding, endocrine resistance, and ErbB signalling pathway (Figure 6B). 

### 2.7. Verification of ADAM17 Expression and Its Prognostic Roles in HCC

The expression of ADAM17 was assessed using IHC in 79 pairs of HCC and adjacent non-tumour tissues. The results showed that the expression of ADAM17 was significantly stronger in HCC tissue compared to those in non-tumour tissues (Figure 7A). Patients with HCC were divided into high and low ADAM17 groups based on the IHC staining intensity. ADAM17 was found to have a close correlation with the clinicopathologic features, including differentiation (*p* = 0.001), histologic grade (*p* = 0.001), stage (*p* = 0.021), and PT (*p* = 0.038) (Table 2). Survival analysis showed that high ADAM17 expression was significantly associated with poorer DFS in patients with HCC (Figure 7B). Univariate analysis of the important factors affecting survival was then performed. The results showed that the length of survival was significantly associated with Child-Pugh (*p* = 0.005), group high expression (*p* = 0), Histologic Grade Gx (*p* = 0.011), PT (*p* = 0.002), and stage (*p* = 0.029) (Table 3). These factors were then entered into the multivariate Cox proportional hazard regression model, and the results demonstrated that group high expression levels and Child-Pugh were independent factors affecting the prognosis of patients with HCC, whereas Histologic Grade Gx, PT, and stage were not (Table 3).

### 2.8. In Vitro Roles of ADAM17 in HCC Cell Growth

To explore ADAM17’s function in HCC, we generated stable ADAM17 knockdown (SK-HEP-1 and MHCC97-H) HCC cells and verified ADAM17 expression using Western blotting (Figure 8A). In the established cell models, ADAM17 knockdown resulted in decreased cell viability, as depicted in Figure 8B. Moreover, inhibition of ADAM17 expression through knockdown hindered clone formation, as illustrated in Figure 8C.

### 2.9. In Vitro Roles of ADAM17 in HCC Metastasis

To determine the effect of ADAM17 on cell metastasis, we performed wound healing and Transwell assays. The results showed that the migration and invasion abilities of ADAM17 knockdown cells were decreased compared to those of the control (Figure 9A–D).

### 2.10. In Vivo Investigation of the Role of ADAM17 in HCC Growth

We explored the oncogenic effects of ADAM17 in a subcutaneously implanted tumour model. The results showed that ADAM17 knockdown significantly suppressed the increase in tumour volume and weight (Figure 10A,B). ADAM17 expression in the tumours was confirmed by immunohistochemistry (IHC; Figure 10C). Ki-67 staining revealed that ADAM17 knockdown decreased the proliferative ability of tumour cells (Figure 10C). Haematoxylin and eosin staining was performed on the collected tumour samples, and the results are presented in Figure 10C.

### 2.11. Verification of the Correlation between ADAM17 and Immune Cell Infiltration

IHC staining was used for validating the association between ADAM17 and immune cell levels in the serial section of HCC tissues. A typical example showing the expression of ADAM17 and immune cell markers (CD3, CD4, CD8, CD56, CD68, CD163, and CD274) is presented in Figure 11A. Quantitative analysis showed that ADAM17 was positively correlated with the immune cell markers (Figure 11B,C; Appendix A), which further verified the roles of ADAM17 in regulating the immune cell infiltration in HCC.

## 3. Discussion

HCC is a malignancy of the digestive system. Many strategies have been attempted to delay disease progression and improve patient prognosis, such as the prevention of chronic liver disease, effective systemic and local treatment, and timely surgical intervention [25]. Although significant progress has been made in the diagnosis and treatment of HCC, its 5-year survival rate is still unsatisfactory [26]. It is, therefore, necessary to identify the possible molecular mechanisms involved and new potential therapeutic targets for HCC. As a transmembrane protein, ADAM17 drives the proteolysis of a variety of chemokines, cytokines, adhesion molecules, and their receptors on the cell membrane [12]. The substrates of ADAM17 include the epidermal growth factor receptor (EGFR) family, IL-6R, Notch1, and tumour necrosis factor alpha (TNFα). These substrates can promote the proliferation, invasion, and migration of tumour cells [12]. In addition, the immunomodulatory effect of ADAM17 on tumour development has attracted attention in recent years [27,28].

In this study, using TCGA and GTEx datasets, we examined the high expression levels of ADAM17 in various tumours, particularly those in HCC tissues, and verified these results using IHC. The Kaplan–Meier method was used to analyse the prognostic role of ADAM17 in HCC, and the results showed that high ADAM17 expression in HCC was associated with a poor prognosis. The relationship between ADAM17 expression and the clinicopathological features of HCC indicates that high ADAM17 expression is associated with poor T and N stages and AFP levels. The function and enriched signalling pathways of ADAM17 in HCC were investigated using GSEA, GO, and KEGG. EMT, inflammatory response, Hedgehog, and KRAS signalling were enriched in patients with high ADAM17 expression. Analysis of the relationship between ADAM17 and immune cell infiltration in HCC using the TIMER database showed that ADAM17 positively correlated with immune cell infiltration and immune checkpoint expression, as verified by IHC. Analysis of the ADAM17 gene profile using the cBioPortal for Cancer Genomics platform showed that ADAM17 has a relatively high mutation rate. PPI network analysis using STRING indicated that ADAM17 plays an important role in tumour progression and immune evasion. Moreover, a series of in vitro and in vivo experiments confirmed that ADAM17 promotes the proliferation, migration, and invasion of HCC cells. Therefore, ADAM17 is highly expressed in HCC and is associated with poor prognosis. Moreover, ADAM17 promoted HCC progression and played a vital role in immune evasion. 

Saad et al. reported that the specific blockade of ADAM17 inhibited cellular proliferation in Kras^G12D^-driven lung adenocarcinoma (LAC) models. Owing to the threonine phosphorylation and preferential shedding of IL-6R, ADAM17 drives IL-6 signalling through the ERK1/2 MAPK pathway, thereby promoting LAC progression. Therefore, they concluded that ADAM17 may be a potential therapeutic target for KRAS-driven LAC [14]. Hedemann et al. demonstrated that the ADAM17 inhibitor, GW280264X, combined with cisplatin significantly improved the treatment of ovarian cancer in two- and three-dimensional models. Therefore, ADAM17 inhibition is considered a promising therapeutic strategy for ovarian cancer [29]. Previous studies have shown that ADAM17 is required for the development of pancreatic ductal adenocarcinoma (PDAC) in mice. Ye et al. found that A9(B8) IgG, an ADAM17 inhibitor, can effectively suppress ADAM17 substrate shedding. Subsequent studies have shown that A9(B8) IgG administration inhibits tumour cell migration and significantly delays PDAC progression in mouse models. Taken together, these results provide an ideal reference for the clinical treatment of PDAC [30]. Bolik et al. demonstrated that the ADAM17 protease contributes to endothelial cell death, tumour cell evasion, and metastasis in tumour necrosis factor receptor 1 (TNFR1)-dependent tumour cells. Moreover, they found that increased γ-secretase release and ADAM17-mediated shedding of TNFR1 ectodomains led to TNF-induced necroptosis. The genetic ablation and pharmacological intervention of ADAM17 in endothelial cells can prevent the rapid formation of metastatic lesions. The data presented by Bolik et al. revealed that ADAM17 may serve as a novel target in advanced malignancies [31].

Macrophages in the TME have an important impact on tumour development. Gnosa et al. reported that the loss of ADAM17 in cancer cell lines leads to a decrease in the expression levels of multiple tumorigenic markers on the surface of co-cultured macrophages in vitro and in mouse models. Owing to the action of ADAM17^−/−^-educated macrophages, the invasive ability of cancer cells is reduced. Further experimental results indicate that the ADAM17 shedding heparin-binding epidermal growth factor (HB-EGF) and amphiregulin in tumour cells are molecular mediators of macrophage education. The HB-EGF ligand can induce the release of chemokine ligands (CXCL) from macrophages, thereby promoting tumour cell invasion [24]. The Fc receptor CD16 is widely distributed in natural killer cells (NK) in the peripheral blood. Romee et al. reported that after NK cell activation, CD16 expression was reduced because of the combinatory effect of cytokines and target cells. ADAM17 protease is expressed by NK cells. When ADAM17 is inhibited, CD16 shedding is reduced, and NK cells enhance the therapeutic efficacy of monoclonal antibodies by increasing cytokine production. Thus, ADAM17 has a substantial impact on NK cell regulation [32]. Programmed cell death ligand 1 (PD-L1) is highly expressed in triple-negative breast cancer cells. Romero et al. found that as members of the ADAM family, ADAM10 and ADAM17 proteases mediate the cleavage of PD-L1. Treating breast cancer cells with the activators of both proteases significantly increased the release of soluble PD-L1 into the culture medium. They speculated that ADAM10 and/or ADAM17 may be involved in regulating the PD-L1/PD-1 pathway and play an important role in antitumour immunity in breast cancer [33].

This study still has several limitations. First, the number of clinical samples is insufficient, which may not be representative of the entire population of patients with HCC. Additionally, there is a lack of in-depth mechanism discussion. However, we are confident that the number of clinical samples will continue to expand with time, and we will also continue to explore the mechanism to strengthen the present conclusions. 

## 4. Materials and Methods

### 4.1. Datasets and Data Preprocessing

Datasets of 33 tumour types, including RNA-sequencing data, mutation data, and patient clinical information, were downloaded from TCGA database https://portal.gdc.cancer.gov (25 August 2023). In addition, expression profile data of normal human tissues were acquired from the University of California, Santa Cruz (UCSC) Cancer Genome Browser https://xenabrowser.net/datapages/ (26 August 2023) The TCGA-LIHC dataset (*n* = 424) was selected for HCC analysis. TPM was used for data normalisation, and data were transformed using log2 (value + 1) before analysis.

### 4.2. ADAM17 Association with Patient Survival and Clinicopathological Features

Patients were divided into two groups based on their ADAM17 expression levels, and patient survival was compared using survival curves. Additionally, we evaluated the role of ADAM17 in the clinicopathological features of patients. T, N, and M stages and AFP levels were compared among patients with distinct ADAM17 expression profiles.

### 4.3. Enrichment Analysis

GSEA was performed to explore the ADAM17-related pathways. Patients with high ADAM17 expression levels were compared with those with low ADAM17 expression, and the HALLMARK gene set was used as the reference gene set for GSEA. The top significant pathways were selected and visualised using the ggplot2 R package [34]. In addition, the DEGs between the low and high ADAM17 expression groups were identified using the DESeq2 platform in R software (version 4.0.2), and a volcano plot was drawn. GO and KEGG analyses were performed based on the DEGs. Top GO annotations and KEGG pathways were displayed using bubble plots.

### 4.4. Immune Cell Infiltration Analysis

The levels of the six immune cell types were estimated for each patient using the TIMER algorithm (https://cistrome.shinyapps.io/timer/ (26 August 2023) [35]. The correlation between ADAM17 expression and the infiltration of each immune cell type was evaluated and visualised using the online TIMER tool. In addition, differences in the abundance of each type of immune cell were identified among patients grouped by variations in ADAM17 copy number.

### 4.5. Mutation Analysis of ADAM17

Genomic ADAM17 mutation datasets were downloaded and analysed using the cBioPortal for Cancer Genomics (http://www.cbioportal.org/ (29 August 2023). The mutation frequency of ADAM17 in pan-cancers, especially in liver cancer, was analysed.

### 4.6. PPI Network Analysis

The top 100 genes directly interacting with ADAM17 were obtained from the STRING database, and a PPI network was established using these genes. GO and KEGG analyses were performed using the genes in this network to clarify the key function of the ADAM17-related network.

### 4.7. Patients and Tumour Samples

We included 79 patients with HCC from June 2020 to December 2021 in Lanzhou University Second Hospital. The inclusion criteria were as follows: (1) patients receiving radical surgery and diagnosed with HCC by histological evaluation; (2) naïve patients without previous antitumour treatment; and (3) those with at least one evaluable lesion. Patients were excluded if they had poor conditions, multiple primary cancer, or unavailable information. The HCC specimens were collected from each included patient to conduct an immunohistochemical assessment. Information on clinicopathological parameters was collected from medical records, and patients were followed up regularly. DFS was evaluated from the date of surgery to the recurrence of tumour or death. The clinical significance of ADAM17 expression was explored on the basis of the included patients. This study was approved by the Institutional Ethics Committee of Lanzhou University Second Hospital (2022A-715). 

### 4.8. Immunohistochemistry

Briefly, sections were baked at 65 °C for 1 h, followed by deparaffinisation, dehydration, and rehydration. After antigen retrieval, the sections were treated with a hydrogen peroxide blocking solution (Maixin, Fuzhou, China). Subsequently, the sections were incubated with primary antibodies (ADAM17, 1:250, Abcam, Cambridge, UK; CD3, CD4, CD8, CD56, CD68, CD163, ready-to-use, Maixin; CD274, 1:100, Invitrogen, Waltham, MA, USA; Ki-67, 1:150, Immunoway, Suzhou, China) at 4 °C overnight and probed with secondary antibodies (ready-to-use, Maixin). Finally, the tissue sections were stained with 3,3′-diaminobenzidine (Maixin), counterstained with haematoxylin (servicebio, Wuhan, China), dehydrated, washed, and mounted.

Immunostaining was evaluated and scored according to the staining intensity and the proportion of stained cells by two independent pathologists who were blinded to the pathological and clinical data. Staining intensity scores were graded as follows: 0 (no staining), 1 (light yellow), 2 (light brown), or 3 (brown). The proportion of stained cells was graded as 0 (<5%), 1 (5–25%), 2 (26–50%), 3 (51–75%), or 4 (>75%). The staining results were calculated using intensity and proportion scores. A staining result ≥4 was considered to reflect a high expression of ADAM17, while that <4 was considered to reflect a low expression of ADAM17. In addition, Image J software was used to quantitatively measure the percentage of positive area of the indicated genes. Dot plots were drawn to explore the correlation between the expression of ADAM17 and the immune cell marker, and Pearson’s correlation coefficient with corresponding P value was calculated. 

### 4.9. Cell Culture and Stable Cell Construction

SK-HEP-1 and MHCC97-H cells were purchased from Shanghai Fuheng Biotechnology (Shanghai, China). The cells were cultured in Dulbecco’s modified eagle medium (DMEM; Gibco, Carlsbad, CA, USA) supplemented with 10% foetal bovine serum (FBS; Gibco). We knocked down the ADAM17 gene by treating liver cancer cell lines with short hairpin RNA (shRNA; Genepharma, Shanghai, China). Cells (1 × 10^5^) were plated and incubated in a six-well plate for 24 h in DMEM supplemented with 10% FBS. Subsequently, cells were transfected with three lentiviral shRNA sequences and one lentiviral control sequence in medium supplemented with 10 μg/mL polybrene (Genepharma). After 16 h of cell transfection, the transfection medium was replaced with fresh medium. ADAM17 stable knockdown cells were achieved by puromycin (SK-HEP-1, 1 μg/mL, MHCC97-H, 2 μg/mL, Biosharp, Guangzhou, China) selection. Verification of knockdown efficiency needs to be performed via Western blotting.

### 4.10. Western Blotting

Control and treated liver cancer cells were harvested and lysed using radioimmunoprecipitation assay (RIPA; Beyotime, Shanghai, China) buffer supplemented with protease inhibitors (Beyotime). After centrifugation (Himac, Tokyo, Japan) at 14,000 rpm (18,700× *g*) for 20 min, the supernatant was carefully transferred to the collection tube. The blots were transferred onto polyvinylidene fluoride (PVDF; Millipore, Danvers, MA, USA) membranes after gel electrophoresis. Non-specific binding sites were blocked with skim milk at room temperature for 1 h. The membranes were incubated in primary antibodies (ADAM17, 1:1000, Abcam, Cambridge, UK; GAPDH, 1:5000, Proteintech, Wuhan, China) overnight at 4 °C. The membranes were then incubated with the appropriate secondary antibodies (anti-mouse/anti-rabbit, 1:5000, Proteintech) at room temperature for 1 h. Signals were observed using the Biospectrum Imaging System (Jiapeng, Shanghai, China).

### 4.11. Cell Proliferation and Colony Formation Assays

Cell proliferation assays were conducted using the Cell Counting Kit 8 (CCK8; Biosharp, China) according to the manufacturer’s instructions. Briefly, cells in the logarithmic growth phase were seeded into 96-well plates at a density of 5000 cells/well. CCK8 reagent was incubated with the cells for 2 h. The cell proliferation status (OD450) was measured at 24, 48, and 72 h. For the cell colony formation assay, cells were seeded into six-well plates at a density of 2000 cells per well and cultured in DMEM supplemented with 10% FBS. After incubation, cell colonies were fixed with 4% paraformaldehyde (servicebio) and stained with 0.1% crystal violet (Solarbio, Beijing, China). The number of colonies was then counted.

### 4.12. Wound Healing Assays

Liver cancer cell lines (Shanghai Fuheng Biotechnology) were seeded into six-well plates. When the cells reached 90–100% confluence, scratch wounds were made using a sterile plastic pipette tip. After washing with PBS, the cells were incubated in a serum-free medium for 24 or 48 h. Scratches were photographed using a phase-contrast microscope (Olympus, Tokyo, Japan).

### 4.13. Transwell Matrigel Invasion and Migration Assay

A Transwell system (BD Biosciences, Franklin Lakes, NJ, USA) was used in this study. In the invasion assay, liver cancer cells (2 × 10^5^ cells/well) in 200 μL of serum-free DMEM were seeded into upper chambers that were previously precoated with diluted Matrigel (Corning, New York, NY, USA). The upper chambers were placed in the lower chamber containing 750 μL of complete cell culture medium. After 24 h of incubation, the upper chambers were immersed in 4% paraformaldehyde for 30 min at room temperature and stained with crystal violet (Solarbio) for 30 min at room temperature. Cells inside the chambers were removed using a cotton swab. The cells at the bottom of the chambers were counted under a microscope (Olympus). The migration assay was performed using the same protocol but without Matrigel, and the number of cells in the chamber was half that of the previous one.

### 4.14. In Vivo Growth Assays

Animal experiments were conducted at the Animal Experiment Center of Lanzhou University Second Hospital and were approved by the Animal Ethics Committee of Lanzhou University Second Hospital (D2022-432). MHCC97-H (2 × 10^6^) cells (Shanghai Fuheng Biotechnology) carrying sh-ADAM17 or control constructs were resuspended in PBS to a 100 μL mixture and were subcutaneously injected into twelve NOD-SCID mice aged 5–6 weeks (GemPharmatech, Nanjing, China). Prior to sample collection, mice were anaesthetised using isoflurane inhalation and euthanised by cervical dislocation. In order to achieve optimal exposure of the surgical field, the tumour was meticulously assessed, revealing partial adhesion to the dorsal fascia and subcutaneous fascia. Employing a combination of blunt and sharp dissection techniques, the tumour was systematically excised, ensuring a thorough and complete removal.

### 4.15. Statistical Analyses

All statistical analyses were performed using the R software (v4.0.2) and GraphPad Prism (v8.0.1). The difference between groups was tested using the Student’s *t*-, Wilcoxon rank sum, chi-squared, or Fisher’s exact tests according to the type of variables. Survival curves were drawn using the Kaplan–Meier method, and differences between the two groups were tested using the log-rank test. All tests were two-tailed, and a *p* value < 0.05 was considered statistically significant.

## 5. Conclusions and Future Directions

In summary, our study establishes ADAM17 as a crucial factor in HCC, influencing tumour development and patient outcomes. High ADAM17 levels correlate with advanced disease stages and poorer survival. Molecular analyses reveal its involvement in key pathways, and it is notably associated with immune responses in the tumour microenvironment. This highlights ADAM17’s potential as a prognostic marker and therapeutic target in HCC.

## Figures and Tables

**Figure 1 ijms-24-17069-f001:**
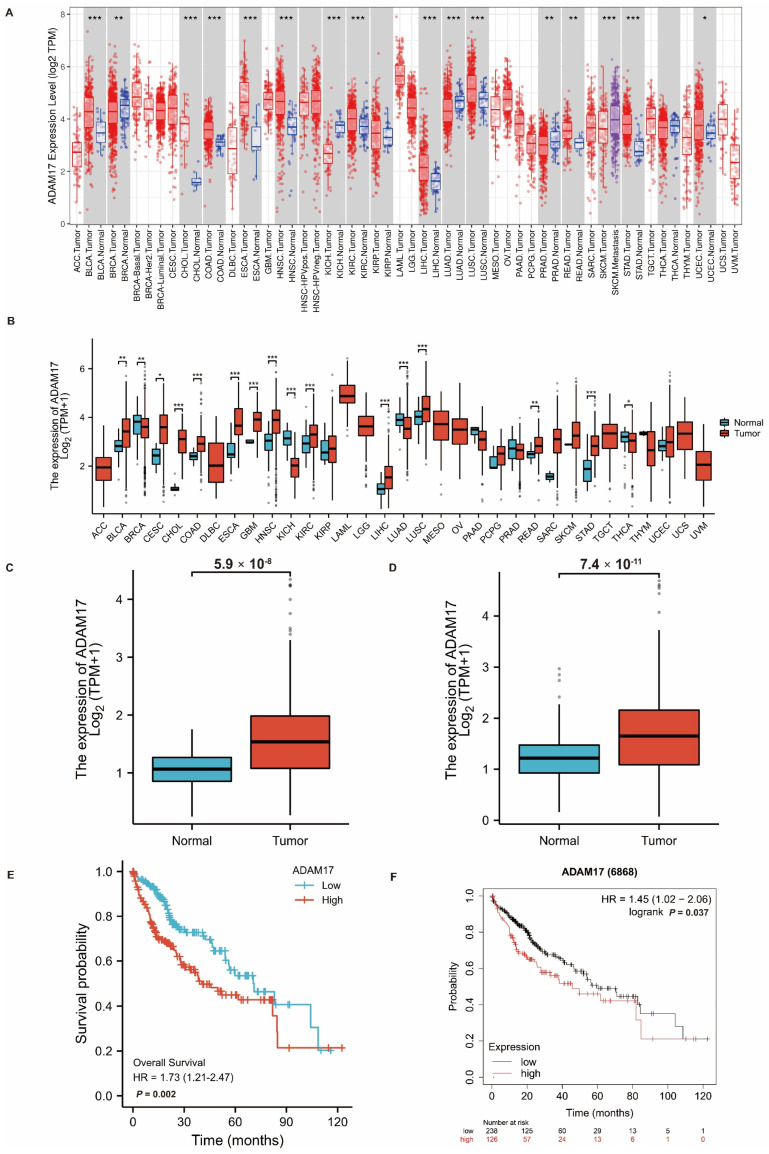
ADAM17 expression in pan-cancer and hepatocellular carcinoma (HCC) and its prognostic role. TIMER (**A**) and TCGA (**B**) databases indicate ADAM17 is upregulated in most cancer types. The expression level of ADAM17 is higher in tumor than normal tissue in TCGA-liver cohorts with ((**C**), *n* = 424) or without ((**D**), *n* = 531) the GTEx datasets. The TCGA ((**E**), *n* = 424) and Kaplan-Meier Plotter ((**F**), *n* = 364) databases show that high expression of ADAM17 is associated with significantly poorer overall survival (OS) in patients with HCC. * *p* < 0.05; ** *p* < 0.01; *** *p* < 0.001.

**Figure 2 ijms-24-17069-f002:**
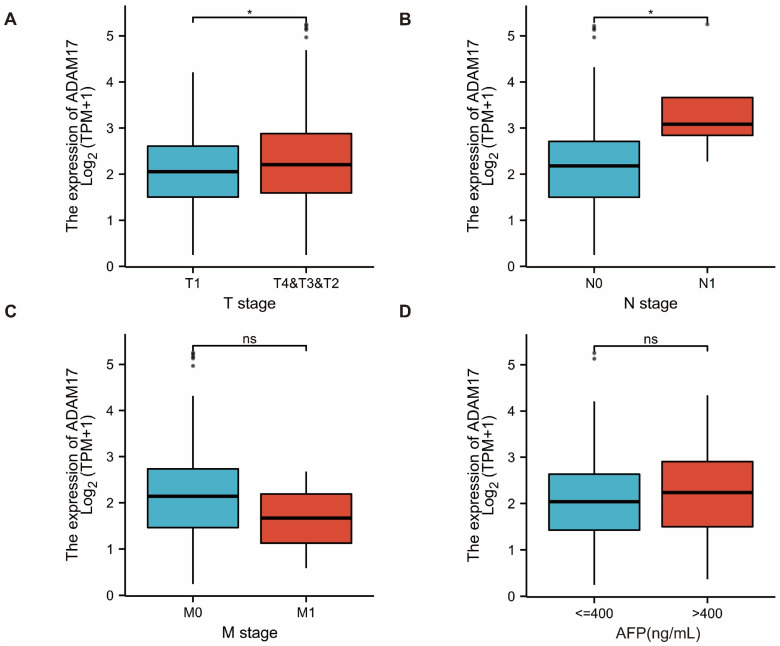
Correlation between ADAM17 levels and the HCC clinicopathological features. Increased ADAM17 expression correlates with more advanced T (**A**) and N (**B**), M stages (**C**), and a higher level of alpha-fetoprotein ((**D**), AFP). ns, no significance; * *p* < 0.05.

**Figure 3 ijms-24-17069-f003:**
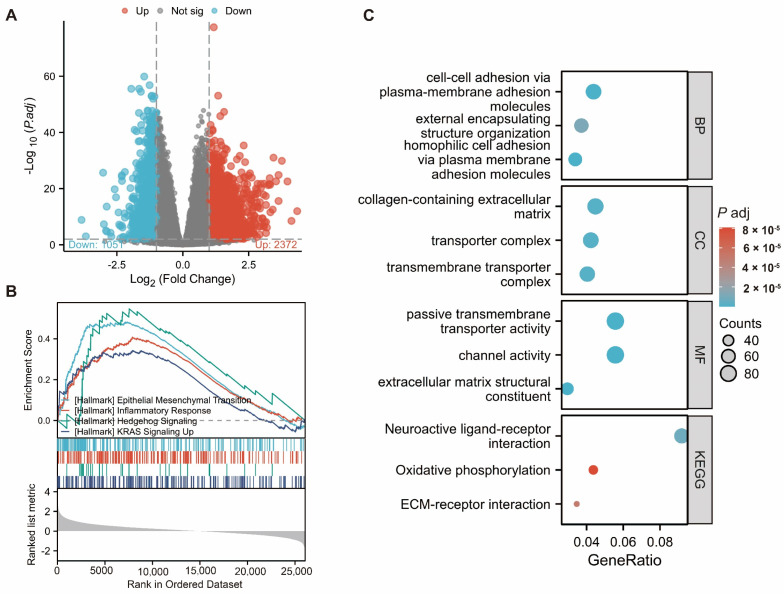
Differentially expressed genes (DEGs) and enrichment analyses. (**A**) Volcano plot showing the DEGs from the analysis by dividing patients into high and low ADAM17 expression groups (*n* = 424); (**B**,**C**) Gene Set Enrichment Analysis (GSEA), Gene Ontology (GO), and Kyoto Encyclopaedia of Genes and Genomes (KEGG) analysis showing the top functions or pathways enriched by the DEGs.

**Figure 4 ijms-24-17069-f004:**
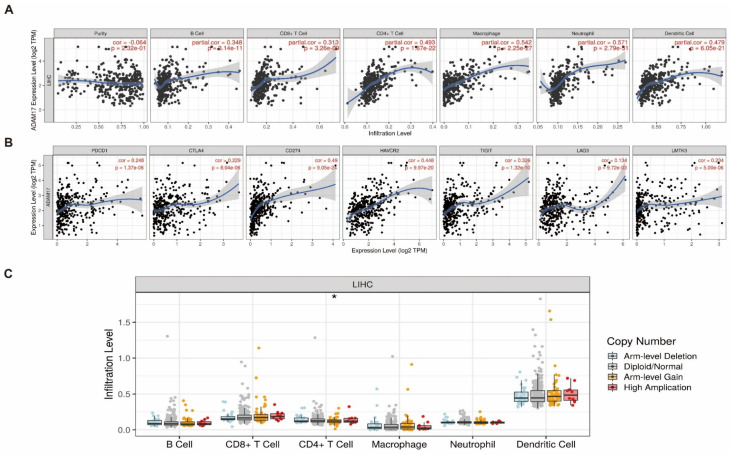
Analysis of immune cell infiltration based on the expression levels of ADAM17. (**A**,**B**) ADAM17 expression levels positively correlated with the infiltration of immune cells and the expression of immune checkpoints; (**C**) the boxplot demonstrates the impact that the ADAM17 copy number has on the immune cell infiltration level. * *p* < 0.05.

**Figure 5 ijms-24-17069-f005:**
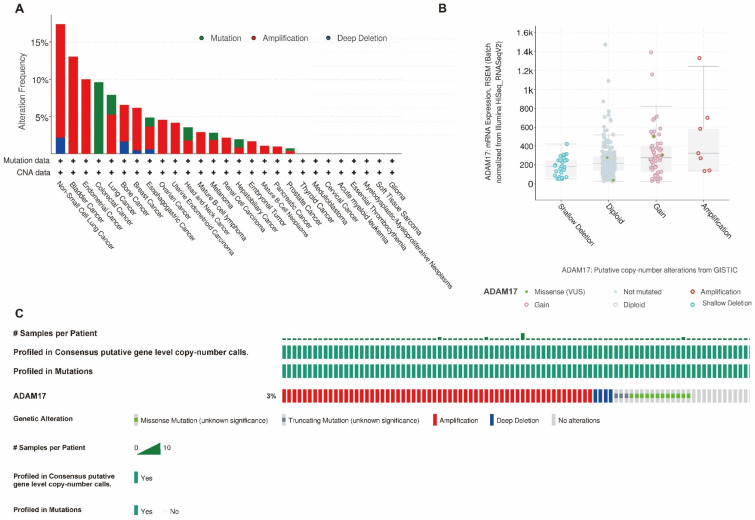
Gene profile analysis of ADAM17 in pan-cancer and HCC. (**A**,**C**) The gene alteration rate in ADAM17 is relatively high in pan-cancer and HCC; (**B**) alterations in ADAM17 copy number correlates with ADAM17 mRNA expression.

**Figure 6 ijms-24-17069-f006:**
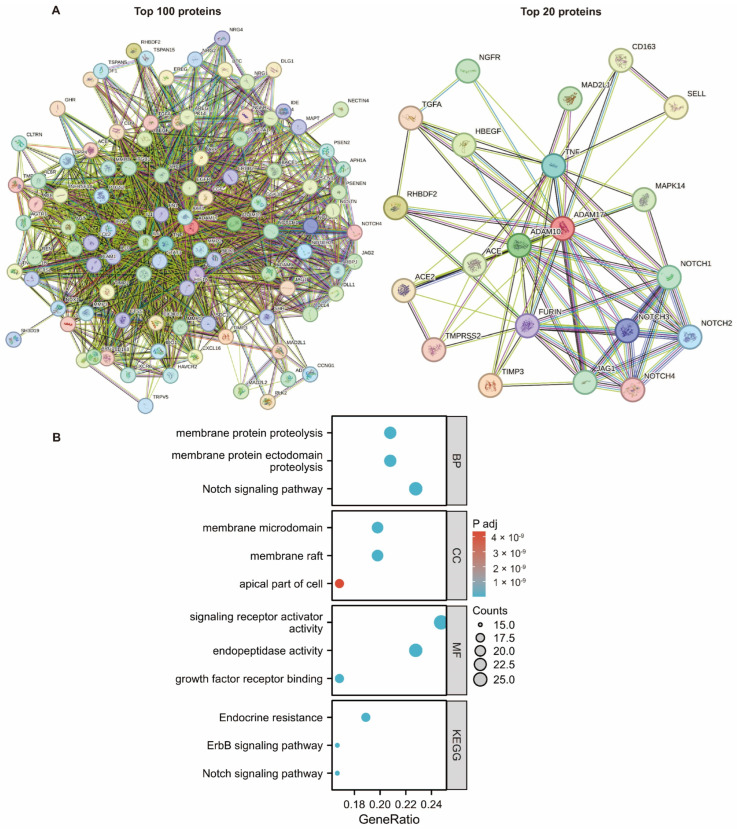
PPI network analysis based on the ADAM17 expression levels. (**A**) Network showing the top proteins directly interacting with ADAM17; (**B**) GO and KEGG pathway analyses based on protein interactions (top 100 proteins).

**Figure 7 ijms-24-17069-f007:**
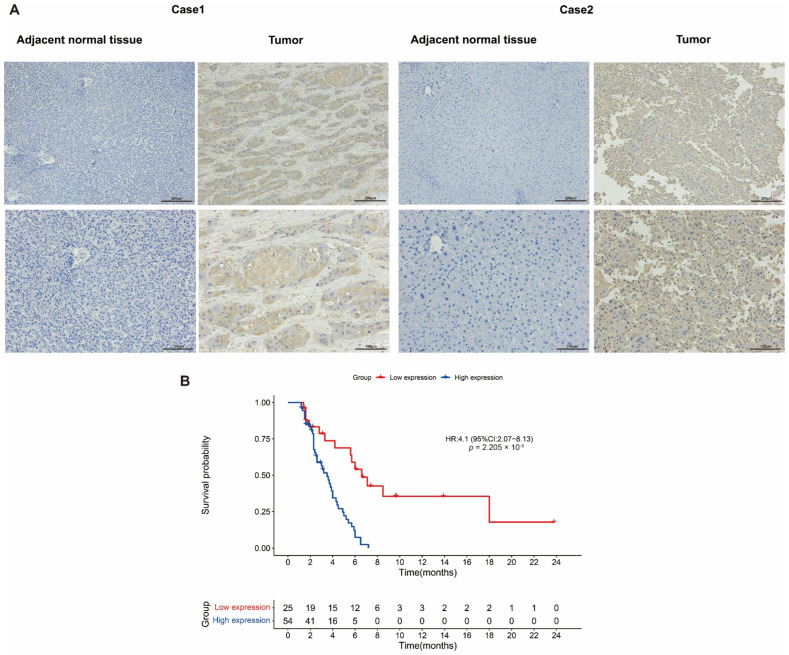
Expression of ADAM17 in real-world HCC patients and its association with patient survival. (**A**) Representative images of IHC staining in tumour and non-tumour tissues. (**B**) Comparison of DFS between patients with high and low ADAM17 expression in 79 HCC patients.

**Figure 8 ijms-24-17069-f008:**
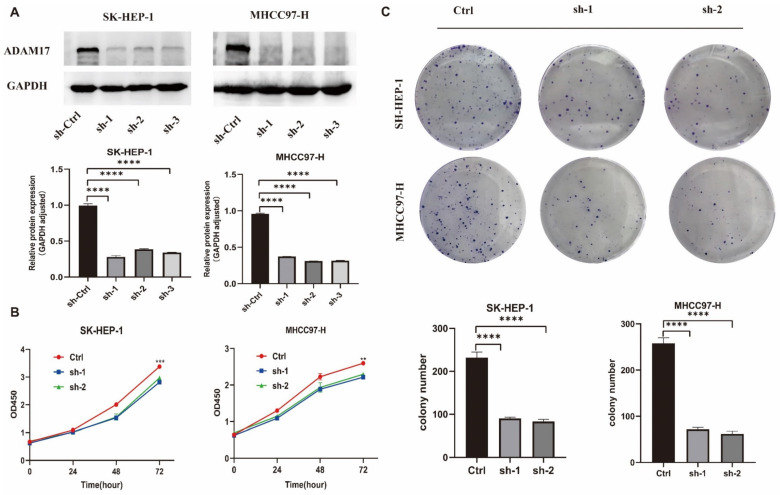
The roles of ADAM17 in promoting HCC cell growth. (**A**) Verification of ADAM17 knock-down in two cell lines; (**B**) cell proliferation of indicated stable cells measured by CCK8; (**C**) the effects of ADAM17 knockdown on cell colony formation. ** *p* < 0.01; *** *p* < 0.001; **** *p* < 0.0001.

**Figure 9 ijms-24-17069-f009:**
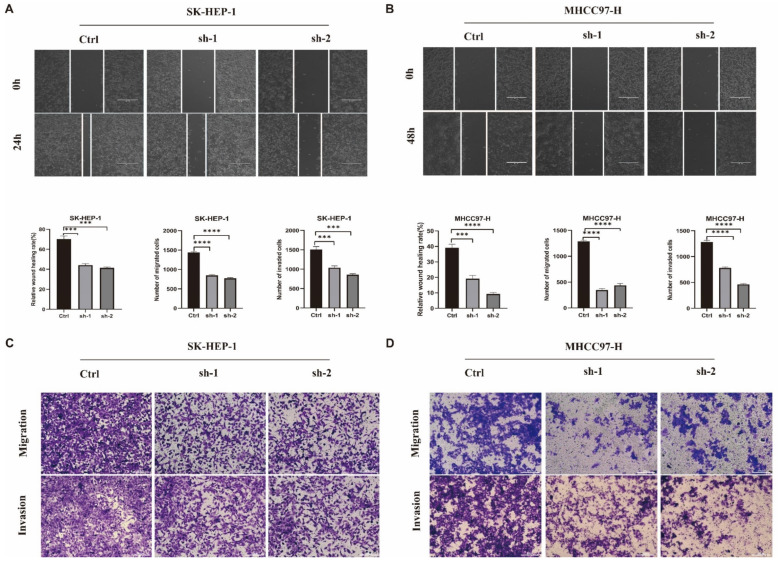
The roles of ADAM17 in promoting HCC cell metastasis. (**A**,**B**) Knockdown of ADAM17 significantly decreased the rates of scratch healing in SK-HEP-1 and MHCC97-H cell lines; (**C**,**D**) knockdown of ADAM17 inhibited cell migration and invasion in the indicated cell lines. *** *p* < 0.001; **** *p* < 0.0001.

**Figure 10 ijms-24-17069-f010:**
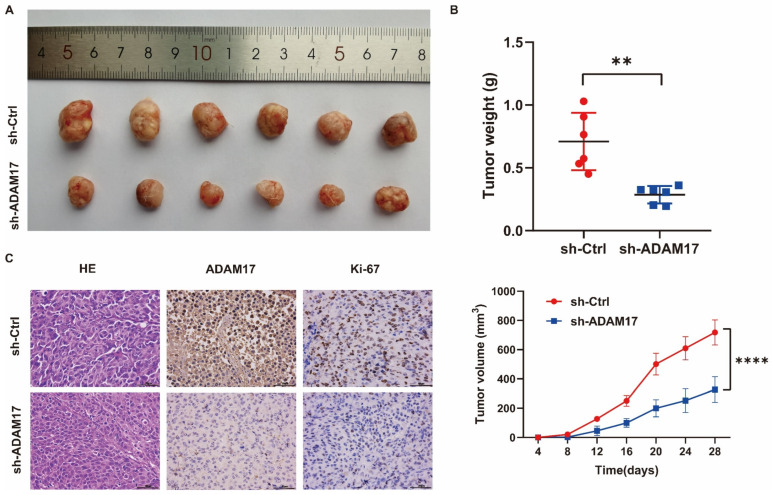
ADAM17 promotes the tumorigenicity in NOD-SCID mice. (**A**) The samples of xenograft tumours harvested from mice (*n* = 12) in different groups 28 days post-inoculation. (**B**) Comparisons of weight and volume among different groups. (**C**) HE staining of indicated subcutaneous xenografts and the expression of Ki-67 and ADAM17 showing by IHC staining. ** *p* < 0.001; **** *p* < 0.0001.

**Figure 11 ijms-24-17069-f011:**
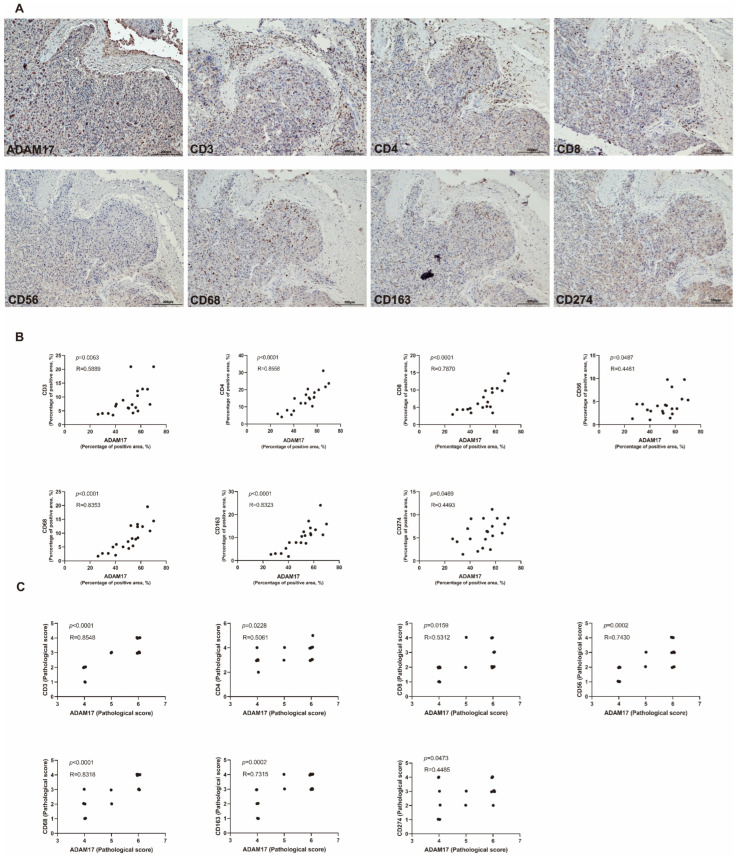
Verification for the immunoregulatory roles of ADAM17 in HCC. (**A**) IHC showing the expression of ADAM17 and immune cell markers in one representative case. (**B**,**C**) Correlation analyses of the association between ADAM17 and the indicated cell markers using the ImageJ quantitative measurement and the pathological score (*n* = 20).

**Table 1 ijms-24-17069-t001:** Associations between ADAM17 expression and the clinicopathological features of HCC patients.

Characteristics	Overall ^$^	Characteristics	Overall
Pathologic T stage, n (%)		Gender, n (%)	
T1	183 (49.3%)	Female	121 (32.4%)
T2	95 (25.6%)	Male	253 (67.6%)
T3	80 (21.6%)	Age, n (%)	
T4	13 (3.5%)	≤60	177 (47.5%)
Pathologic N stage, n (%)		>60	196 (52.5%)
N0	254 (98.4%)	BMI, n (%)	
N1	4 (1.6%)	≤25	177 (52.5%)
Pathologic M stage, n (%)		>25	160 (47.5%)
M0	268 (98.5%)	Histologic grade, n (%)	
M1	4 (1.5%)	G1	55 (14.9%)
Pathologic stage, n (%)		G2	178 (48.2%)
Stage I	173 (49.4%)	G3	124 (33.6%)
Stage II	87 (24.9%)	G4	12 (3.3%)
Stage III	85 (24.3%)	AFP (ng/mL), n (%)	
Stage IV	5 (1.4%)	≤400	215 (76.8%)
		>400	65 (23.2%)

^$^: the ‘overall’ column shows the sample size with corresponding percentage for each characteristic category.

**Table 2 ijms-24-17069-t002:** Relationship between ADAM17 expression and clinicopathological parameters of patients with HCC.

Clinicopathological Parameters	Level	ADAM17 Expression		*p*-Value
		Low Expression	High Expression	
N		25	54	
Age (median [IQR])		51.00 [46.00, 59.00]	56.00 [49.00, 59.75]	0.509
Sex (%)	female	10 (40.0)	12 (22.2)	0.114
	male	15 (60.0)	42 (77.8)	
Cirrhosis (%)	no	2 (8.0)	2 (3.7)	0.587
	yes	23 (92.0)	52 (96.3)	
Capsule (%)	incomplete	2 (8.0)	10 (18.5)	0.458
	intact	6 (24.0)	9 (16.7)	
	unclear	17 (68.0)	35 (64.8)	
MVI risk (%)	M0	14 (56.0)	24 (44.4)	0.678
	M1	9 (36.0)	25 (46.3)	
	M2	2 (8.0)	5 (9.3)	
Differentiation (%)	low	0 (0.0)	8 (14.8)	0.001
	moderate	19 (76.0)	41 (75.9)	
	high	6 (24.0)	1 (1.9)	
	unclear	0 (0.0)	4 (7.4)	
Histologic Grade (%)	G1	6 (24.0)	1 (1.9)	0.001
	G2	19 (76.0)	41 (75.9)	
	G3	0 (0.0)	8 (14.8)	
	Gx	0 (0.0)	4 (7.4)	
Stage (%)	I	12 (48.0)	10 (18.5)	0.021
	II	10 (40.0)	23 (42.6)	
	III	3 (12.0)	18 (33.3)	
	IV	0 (0.0)	3 (5.6)	
History of hepatitis (%)	no	3 (12.0)	3 (5.6)	0.374
	yes	22 (88.0)	51 (94.4)	
TBIL μmol/L (median [IQR])		13.90 [11.80, 21.50]	16.00 [11.60, 24.60]	0.602
ALB g/L (median [IQR])		37.70 [34.70, 38.50]	37.80 [36.20, 41.38]	0.158
PT s (median [IQR])		12.10 [11.20, 12.60]	12.70 [12.00, 13.35]	0.038
Child Pugh (%)	A	24 (96.0)	50 (92.6)	1
	B	1 (4.0)	3 (5.6)	
	C	0 (0.0)	1 (1.9)	
Tumor size (%)	MHCC	4 (16.0)	14 (25.9)	0.702
	SHCC	10 (40.0)	22 (40.7)	
	LHCC	9 (36.0)	14 (25.9)	
	HHCC	2 (8.0)	4 (7.4)	
Serum AFP ng/mL (%)	≤400	19 (76.0)	44 (81.5)	0.563
	>400	6 (24.0)	10 (18.5)	

Note: MVI, microvascular invasion; TBIL, total bilirubin; ALB, serum albumin; PT, prothrombin time; MHCC, micro hepatocellular carcinoma; SHCC, small hepatocellular carcinoma; LHCC, large hepatocellular carcinoma; HHCC, huge hepatocellular carcinoma; AFP, alpha fetoprotein.

**Table 3 ijms-24-17069-t003:** Univariate and multivariate Cox regression survival analysis of clinicopathological parameters and ADAM17 expression in patients with HCC.

Clinicopathological Parameters		Univariate Analysis			Multivariate Analysis		
		HR	95% CI	*p*-Value	HR	95% CI	*p*-Value
Age		1.01	0.98–1.04	0.449	NA	NA	NA
ALB g/L		1	0.92–1.07	0.918	NA	NA	NA
Capsule	Incomplete	Reference					
	intact	0.83	0.34–2.04	0.679	NA	NA	NA
	unclear	0.86	0.42–1.75	0.678	NA	NA	NA
Child Pugh		3.41	1.45–8.02	0.005	3.48	1.09–11.11	0.0356
Cirrhosis	No	Reference					
	yes	5.56	0.77–40.24	0.09	NA	NA	NA
Differentiation	Low	Reference					
	high	0.24	0.05–1.03	0.054	NA	NA	NA
	moderate	0.74	0.29–1.92	0.535	NA	NA	NA
	unclear	1.73	0.46–6.47	0.416	NA	NA	NA
group	Low expression	Reference					
	High expression	4.07	2.06–8.08	0	3	1.4–6.42	0.0047
Histologic Grade	G1	Reference					
	G2	3.13	0.97–10.13	0.057	1.83	0.53–6.37	0.341
	G3	4.22	0.98–18.3	0.054	1.35	0.27–6.88	0.7159
	Gx	7.3	1.59–33.56	0.011	3.44	0.67–17.77	0.1405
History of hepatitis	No	Reference					
	yes	1.39	0.55–3.49	0.487	NA	NA	NA
MVI risk		1.11	0.72–1.7	0.633	NA	NA	NA
PT s		1.31	1.1–1.56	0.002	1.07	0.84–1.35	0.601
Serum AFP ng/mL	≤400	Reference					
	>400	1.37	0.72–2.6	0.332	NA	NA	NA
Sex	Female	Reference					
	male	1.29	0.71–2.33	0.403	NA	NA	NA
Stage		1.45	1.04–2.03	0.029	1.42	0.98–2.05	0.063
TBIL μmol/L		1.02	0.99–1.05	0.162	NA	NA	NA
Tumor size		1.09	0.81–1.46	0.57	NA	NA	NA

Note: NA, not available; HR, hazard ratio; CI, confidence interval; MVI, microvascular invasion; TBIL, total bilirubin; PT, prothrombin time; AFP, alpha fetoprotein.

## Data Availability

All data are contained within the manuscript. The datasets used and/or analysed during the current study are available from the corresponding author on reasonable request.

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
