# Peer review of "Predictive Roles of ADAM17 in Patient Survival and Immune Cell Infiltration in Hepatocellular Carcinoma"

_ijms, 2023, doi:10.3390/ijms242317069_

Round 1
Reviewer 1 Report
Comments and Suggestions for Authors
The enclosed manuscript effectively details the relation between increased expression levels of ADAM17 in patients with hepatocellular carcinoma as a potential biomarker for indicating disease severity. The authors do a superb job of outlining the relevance of examining this gene as a biomarker and the included data demonstrates the comprehensive nature of this study. In particular the examination of expression levels both via RNA expression and in-situ based on immunohistochemistry imaging are powerful examples that support the conclusions drawn in this study. As the work is also highly relevant to the field, it is the opinion of this reviewer that the work will be ready for publication after addressing a few minor points which will help to clarify the presented data for readers. These comments can be found below:
Minor Comments:
- For all tissue section images please ensure that images are white balanced and contrast is appropriately adjusted so as to increase visibility of stain for readers. In particular an example of this can be found in Figure 11 in which the IHC images are a bit too dark for readers to easily distinguish between the background hematoxylin counterstain and the HRP staining for the various markers.
- Additionally, for quantitative measurement of IHC images the methods for generating the dot plots in Figure 11 was not expressly detailed. The methods describe the development of a categorical scale for quantifying images but the coupling of these data point on the 0-4 point scale with the GEPIA database requires additional detail. Furthermore, as these images are being compared for similar locations the application of readily available image analysis software such as ImageJ packages may offer a more objective percentage stain per image value when grading these samples. The application of both scores generated by software and independent pathologists could further bolster the demonstrated results.
- For associated genes and pathways derived from String database and presented in Figure 6, it may be beneficial to reduce the number of presented genes in the network. While it is interesting to show the 100 genes currently presented in the figure it makes the diagram difficult for readers to glean meaningful information from as the connections shown overlap to too great of a degree. By filtering this nodal relation map to 10-20 key gene targets it will dramatically improve readability and be more impactful for readers interested in exploring associated genes based on this work.
- Several figure, such as Figure 1 and 2, could benefit from more descriptive captions that detail what is being shown in each sub-set. For example, in Figure 1 sub-set plot A and B appear to show the same information, this is the same case for sub-set plots E and F, so it would be helpful for the caption of this figure to highlight the individual sub-set plots to better describe what the differences are. If instead these sub-set plots are showing the same data just in different visual formats, then it would be recommended to select only one format so that the exact same data is not redundantly presented.
- In Table 1, please clarify in caption what is being demonstrated by the "overall" column as this column title by itself is non-descript.
- In line 98, the term "hallmark" is shown in all caps is there a reason that this is fully capitalized, if not please adjust?
- For section 4.9 please provide additional detail for transfection parameters as these parameters can be important information for reproducibility.
- For section 4.10 please either provide rotor arm length or units in RCF for reproducibility.
- For section 4.14 please provide further detail on the extraction and comparative process for extracted tumors. Specifically please detail how tumor boundaries for excision were determined.
- The conclusion section, while it does appear to be supported by the presented results would benefit from being built out a bit more to further describe the findings in this study. As there are multiple assays being implemented to address this correlation between ADAM17 and HCC severity it would be helpful for readers to have these findings summarized and contextualized in the conclusion in more than a two sentence statement.
Reviewer 2 Report
Comments and Suggestions for Authors
Please see attached document
